# Effects of Treatment with a DNA Methyltransferase Inhibitor 5-aza-dC on Sex Differentiation in Medaka (*Oryzias latipes*)

**DOI:** 10.3390/ijms26073280

**Published:** 2025-04-01

**Authors:** Xiaojuan Cui, Liumeiyang Xu, Nan Tian, Jianjun Peng

**Affiliations:** College of Life Sciences and Health, Hunan University of Science and Technology, Xiangtan 411201, China; xjcui@hnust.edu.cn (X.C.); 15898531806@163.com (L.X.); lalala21123@163.com (N.T.)

**Keywords:** DNA methylation, DNA methyltransferases (DNMTs), sex differentiation, 5-aza-dC, *Oryzias latipes*

## Abstract

DNA methylation is a common epigenetic modification of DNA levels in the genome of eukaryotic cells, and an aberrant elevation of DNA methylation in gene promoter regions can inhibit gene expression. DNA methyltransferases (DNMTs) are involved in genomic DNA methylation, divided into maintenance DNA methyltransferases and de novo methylases, which are expressed to different degrees in the testis and ovaries. 5-aza-2′-deoxycytidine (5-aza-dC) is a cytidine analog with a strong methylation inhibition. In this experiment, medaka fish fries were treated with 5-aza-dC at 0 μg/L, 50 μg/L, and 100 μg/L. It was found that 100 g/L concentration of 5-aza-dC inhibited both body length and body weight of the adult fish, while 50 g/L concentration had no significant difference. In addition, paraffin section observation and gonad index statistics showed that after 100 g/L concentration of 5-aza-dC treatment, the gonad index of female fish increased significantly, but the gonad index of male fish had no significant difference. And the development of sperms and ovaries was normal without significant difference. Finally, we found that 5-aza-dC not only significantly decreased the transcription levels of *dnmt1* and *dnmt3bb.1*, but also significantly increased the expression levels of female-related genes such as *foxl2*, *cyp19a1* and *wnt4*, and significantly decreased the expression levels of male-related genes such as *dmrt1*, *sox9a* and *amh*. The DNA methylation patterns of *foxl2* and *dmrt1* genes were altered. This work provides more references for understanding the mechanism of DNA methylation affecting sex determination in fish.

## 1. Introduction

DNA methylation is an important way to achieve epigenetic modification, which can change the genetic epigenetics without changing the DNA sequence and play a key role in gene expression regulation, genome imprinting, X chromosome inactivation, and other events [1,2]. DNA methylation falls into two main categories: de novo and maintenance methylation, mediated by DNA methyltransferases (DNMTs) such as *dnmt1* (maintenance) and *dnmt3* (de novo) [3]. Research indicates that DNA methylation remodeling is crucial in high-temperature induced masculinization in fish, where it suppresses female-related gene expression. For instance, in zebrafish and eels, the DNMT inhibitor 5-aza-dC can reverse sex and increase female ratios, though the underlying mechanisms remain unclear [4]. Furthermore, DNA methylation has been shown to regulate gonad differentiation in eels during gonad development [5].

DNA methylation patterns in the genome are primarily catalyzed by DNMTs [6]. The currently identified DNMTs include *dnmt1*, *dnmt2*, *dnmt3a*, *dnmt3b*, and *dnmt3l*, where *dnmt1* is the maintenance methyltransferase [7]. Elevated methylation in gene promoter regions typically suppresses gene expression, while DNMT inhibitors like 5-aza-dC can reverse this by demethylating CpG sites where cytosine (C) is followed by guanine (G) [8]. 5-aza-dC, a cytidine analog, specifically targets DNA with minimal effects on RNA or protein synthesis, making it a widely used demethylating agent [9]. In the embryogenesis of Japanese rice fish, 5-aza-dC has the potential to regulate the developmental rhythm and methylation mechanism [5].

Medaka (*Oryzias latipes*), a model organism for genetic studies due to its ease of breeding and short reproductive cycle, has an XX/XY sex determination system. In the Hd-rR strain, the *dmrt1Y* gene on the Y chromosome is the sex determination gene, and the *dmrt1* gene on the autosomes is the sex differentiation gene [10,11,12]. Through the regulation of germ cell formation or estrogen synthesis in the sex, *dmrt1* plays a key role in the development and structural maintenance of the testis. Both the sex steroid hormone and high-temperature induction can alter the sex of medaka to XX male by affecting DNA methylation levels [13,14]. Moreover, the DNA methylation program in medaka embryos is similar to that of human and mouse embryos, making it important to select the medaka as the research object [15].

Sex determination and differentiation in fish involve genes like *dmrt1*, *sox*, *TGF-β*, and *foxl2* and the *wnt* pathway. *Dmrt1*, *gsdf*, and *sox* promote testicular development [16]. Sex differentiation in fish is influenced by genetic and environmental factors such as hormones, temperature, and pH [17]. Temperature, in particular, plays a critical role, with elevated temperatures often inducing male development [18].

0 µg/L, 50 µg/L, and 100 µg/L 5-aza-dC were used to treat medaka to analyze its effects on gonad development, DNMT expression, and DNA methylation [19,20]. The results showed that high concentration of 5-aza-dC significantly affected female gonad development and regulated gene expression by altering DNA methylation patterns. This study provides new evidence that DNA methylation affects sex determination in fish.

## 2. Results

### 2.1. The Effect of Treatment with Different Concentrations of 5-aza-dC on the Growth Indicators of Medaka Fish

We measured the body length of medaka treated with 5-aza-dC at different gradient concentrations, and then analyzed the data. The body lengths of male and female medaka treated with 0 μg/L 5-aza-dC concentration (control group) were 3.05 ± 0.18 cm (*n* = 46) and 3.01 ± 0.07 cm (*n* = 48), respectively. That of male and female medaka treated with 50 μg/L 5-aza-dC were 3.20 ± 0.31 cm (*n* = 36) and 3.00 ± 0.21 cm (*n* = 32), respectively, which showed no significant difference compared with the control group. Especially, the body lengths of male and female medaka treated with 100 μg/L 5-aza-dC were 2.48 ± 0.24 cm (*n* = 37) and 2.75 ± 0.24 cm (*n* = 44), respectively, which were significantly lower than those of the control group (*p* < 0.001, *p* < 0.05). Thus, high concentrations of 5-aza-dC inhibited the growth of the body length of the medaka (see Figure 1a).

In addition to body length, we analyzed the effects of 5-aza-dC on medaka body weight. In the control group (0 μg/L 5-aza-dC), females exhibited a mean body weight of 0.31 ± 0.05 g (n = 46), significantly higher than males (0.26 ± 0.02 g, *n* = 48; *p* < 0.05). Treatment with 50 μg/L 5-aza-dC did not significantly alter body weight in either sex (females: 0.33 ± 0.11 g, *n* = 36; males: 0.27 ± 0.06 g, *n* = 32; *p* > 0.05 vs control). However, exposure to 100 μg/L 5-aza-dC resulted in markedly reduced body weights for both females (0.17 ± 0.05 g, *n* = 37) and males (0.21 ± 0.04 g, *n* = 44), representing significant decreases compared to controls (*p* < 0.01 and *p* < 0.05, respectively). These findings demonstrate that high-concentration 5-aza-dC treatment significantly impairs body weight gain in medaka (see Figure 1b).

### 2.2. Effects of Different Concentrations of 5-aza-dC Treatment on the Gonad Development of Medaka Fish

To assess the effects of 5-aza-dC on gonad development, we calculated the gonad index (gonad weight relative to body weight). In the control group (0 μg/L 5-aza-dC), the gonadal indices of female and male were 3.77 ± 1.60% (*n* = 9) and 1.51 ± 1.06% (*n* = 8), respectively. At 50 μg/L 5-aza-dC, the gonad indices of females and males were 3.75 ± 1.74% (*n* = 9) and 2.14 ± 1.60% (*n* = 8), respectively. However, exposure to 100 μg/L 5-aza-dC induced a significant increase in the female gonad index (6.82 ± 3.53%, *n* = 7; *p* < 0.05 vs. control), while the male gonad index remained unchanged (1.29 ± 0.61%, *n* = 6; *p* > 0.05). These results indicate that high-dose 5-aza-dC selectively enhances gonad development in female medaka without significantly affecting males (see Figure 2).

To characterize germ cell development in medaka, we performed histological examination of gonad paraffin sections. In ovarian tissue across all 5-aza-dC treatment groups, we observed oocytes at all developmental stages (IA, IB, II, III, and IV). Similarly, testicular tissue contained the full complement of spermatogenic cells, including spermatogonia (Sg), spermatocytes (Sc), and sperm (Sp) (see Figure 3).

### 2.3. Sex Determination at the Genetic Level

Genomic DNA extracted from tail fins of medaka exhibiting female secondary sexual characteristics was analyzed using a dual-PCR approach. *dmrt1Y* gene detection: Successfully amplified Y-chromosomal marker in sample No. 2 (Employed sex-specific primers PG19/PG20) (Figure 4a). β-actin (housekeeping gene) amplification confirmed: Successful DNA extraction in all samples (Employed primers β-actin4F/4R) (Figure 4b). The results showed that the DNA of all samples was successfully extracted, and the *dmrt1Y* gene fragment was detected in the sample at sampling hole No. 2, indicating that the sample was genetically male (i.e., XY).

### 2.4. Analysis of Expression Levels of DNA Methyltransferase Gene and Genes Related to Sex Differentiation

#### 2.4.1. DNA Methyltransferase Gene Expression Level Analysis

The level of gene expression was calibrated by an internal reference. The expression level of *dnmt1* was inhibited at 50 μg/L and 100 μg/L 5-aza-dC (*p* < 0.05, *p* < 0.05). The expression level of *dnmt3bb.1* was inhibited at 50 μg/L and 100 μg/L 5-aza-dC (*p* < 0.05, *p* < 0.01). However, there was no significant difference in the expression level of *dnmt3bb.2* at 50 μg/L and 100 μg/L 5-aza-dC (see Figure 5A(a–c)).

#### 2.4.2. Analysis of Expression Levels of Genes Related to Sex Differentiation

The level of gene expression was calibrated by internal reference. Quantitative analysis revealed significant upregulation of key genes involved in female sex differentiation following 5-aza-dC exposure. *foxl2* expression: Significant upregulation at both 50 μg/L (*p* < 0.05) and 100 μg/L (*p* < 0.05); *wnt4* expression: Marked increase at 50 μg/L (*p* < 0.001) and Strong induction at 100 μg/L (*p* < 0.001); *cyp19a1* expression: Significant elevation at both concentrations (*p* < 0.001 for each)(see Figure 5B(a–c)). Quantitative analysis revealed significant alterations in key male sex differentiation markers following 5-aza-dC exposure. *dmrt1* expression: Significant downregulation at 50 μg/L (*p* < 0.01) and Moderate suppression at 100 μg/L (*p* < 0.05); *amh* expression: Dramatic inhibition at both concentrations (*p* < 0.001 for each); *sox9a* expression: Biphasic response with upregulation at 50 μg/L (*p* < 0.05) and Downregulation at 100 μg/L (*p* < 0.05) (see Figure 5B(d–f)).

### 2.5. Methylation Levels of dmrt1 and foxl2 in Response to 5-aza-dC Inhibitors

In accordance with the expression profiles of gender-associated genes, we chose the male-specific gene *dmrt1* and the female-specific gene *foxl2* to investigate their DNA methylation patterns. We measured and analyzed the methylation levels of these two genes following exposure to varying concentrations of 5-aza-dC. From the numerous sites examined, we focused on two specific sites in *dmrt1* and six sites in *foxl2* for detailed evaluation. The results related to CpG dinucleotides indicated that under the influence of 5-aza-dC, the methylation levels at −1996 bp and −1751 bp within the promoter region of *dmrt1* increased. Conversely, the methylation levels at −1944 bp, −1928 bp, −1868 bp, −1848 bp, −1826 bp and −1718 bp within the promoter region of *foxl2* decreased. Further comparisons across different treatment concentrations revealed that at a concentration of 50 μg/L of 5-aza-dC, there was no significant change in the methylation level associated with the *dmrt1* gene. However, when the concentration was elevated to 100 μg/L, the methylation level of the male-related *dmrt1* gene significantly increased (Figure 6A). In contrast, for the *foxl2* gene, both concentrations of 50 μg/L and 100 μg/L of 5-aza-dC led to a reduction in methylation levels. Notably, the decrease in methylation was more pronounced at 100 μg/L compared to 50 μg/L (Figure 6B).

## 3. Discussion

The sex of fish is influenced by environmental and genetic factors. An increasing number of studies have demonstrated that DNA methylation plays a crucial role in the sex determination and differentiation processes of fish [21]. Inhibiting DNA methyltransferases with DNA methyltransferase inhibitors is an effective approach for investigating the functions of DNA methyltransferases.

5-Aza-2′-deoxycytidine (5-aza-dC), a cytosine analogue, is recognized as one of the most effective demethylating agents discovered to date [22]. Upon interaction with DNA methyltransferases, 5-aza-dC becomes methylated and irreversibly binds to these enzymes, forming covalent adducts [23]. This property renders 5-aza-dC crucial for investigating the role of DNA methylation in epigenetics, the expression and regulation of specific genes, and provides a valuable tool for modulating DNA methylation levels to study the impact of gene methylation. 

Currently, studies on some teleost fishes have explored the function of the DNA methyltransferase inhibitor 5-aza-dC in sex differentiation, yielding variable results. For example, research on zebrafish has shown that treatment with 5-aza-dC can induce feminization [24]. In cases where aromatase inhibitors cause female-to-male sex reversal in zebrafish, simultaneous treatment with 5-aza-dC and *aromatase* inhibitors maintains the female phenotype, suggesting that 5-aza-dC can counteract the effects of *aromatase* inhibitors [25]. Additionally, another study demonstrated that high temperature induces masculinization in zebrafish. Co-treatment with 5-aza-dC mitigates this effect by reducing the methylation levels of the *esr1* and *sox9b* promoters, restoring their expression, and consequently decreasing the male-biased sex ratio [26].

Medaka, as a model organism, has emerged as a crucial experimental subject in modern biological research because of its merits like a short sexual maturation cycle, low rearing difficulty, and strong stress resistance [27,28]. Despite the fact that both medaka and zebrafish are key model organisms in the field of fish research, currently, studies on the treatment of medaka with 5-aza-dC are relatively scarce. Only Dasmahapatra et al. treated medaka with 5-azacytidine (5-azaC) and disclosed the mechanism by which DNA methyltransferase is regulated by it [29]. 

In this study, through experimental analysis, we demonstrated that high concentrations of 5-aza-dC significantly influence the sex determination mechanism in medaka fish. Specifically, it not only markedly suppresses the transcriptional levels of *dnmt1* and *dnmt3bb* but also exerts a profound regulatory effect on the expression levels of female- and male-related genes. Notably, *dnmt1* and *dnmt3bb*, as critical DNA methyltransferases, play essential roles in the DNA methylation modification process [30]. The significant reduction in their transcriptional levels caused by high concentrations of 5-aza-dC suggests effective inhibition of the DNA methylation process. This inhibition may trigger subsequent gene expression changes, thereby profoundly impacting the sex determination and differentiation processes in medaka fish.

It is worth emphasizing that high concentrations of 5-aza-dC significantly upregulate the expression of female-related genes such as *foxl2*, *cyp19a1*, and *wnt4*. The *foxl2* gene, a member of the forkhead transcription factor family, plays a pivotal role in the development and maintenance of female gonads across various organisms [31]. In medaka fish, 5-aza-dC likely inhibits DNA methylation by suppressing the methylation status of the *foxl2* promoter region, thereby enhancing its transcriptional activity and increasing its expression level. The *cyp19a1* gene encodes aromatase, which is crucial for estrogen synthesis, converting androgens into estrogens [32]. The *wnt4* gene plays an indispensable role in ovarian development and the maintenance of the female reproductive system [33]. Therefore, the marked increase in the expression levels of these three female-related genes implies that under the influence of 5-aza-dC, estrogen synthesis in medaka fish increases, potentially activating ovarian development-related signaling pathways and promoting the development of female characteristics.

Conversely, high concentrations of 5-aza-dC significantly downregulate the expression of male-related genes such as *dmrt1*, *sox9a*, and *amh*. The *dmrt1* gene is a key regulator of sex determination and differentiation in vertebrates, playing an essential role in the development and maintenance of male germ cells and male characteristics [34]. The *sox9a* gene is vital for testis development, spermatogenesis, and the regulation of testicular determination and development. The *amh* gene is crucial for the normal development and function of the male reproductive system, inhibiting the development of female reproductive duct precursors while promoting the development of the testis and male reproductive system [35]. Thus, the significant reduction in the expression levels of these three male-related genes suggests that high concentrations of 5-aza-dC inhibit the development of the male reproductive system and the activation of associated signaling pathways.

More notably, in this study, high concentrations of 5-aza-dC reduce the methylation level of the female-related gene *foxl2* in medaka fish while significantly increasing the methylation level of the male-related gene *dmrt1*. From an epigenetic perspective, this differential methylation change may imply a more complex regulatory mechanism. On one hand, changes in DNA methylation levels result from the combined action of multiple factors, including DNA sequence characteristics, histone modification status, and chromatin structure [36]. In medaka fish, 5-aza-dC may differentially affect these factors, leading to opposing methylation trends for the *foxl2* and *dmrt1* genes. On the other hand, such differential methylation changes may be closely linked to upstream and downstream gene regulatory networks and intracellular signal transduction pathways [37]. During the sex determination and differentiation process in medaka fish, a complex gene regulatory network exists where various genes interact and regulate each other. 5-aza-dC may indirectly alter the methylation status of the *foxl2* and *dmrt1* genes by influencing the expression or activity of certain key regulatory factors.

In summary, high concentrations of 5-aza-dC exert significant and complex effects on the expression and methylation levels of sex-related genes in medaka fish. These effects involve not only the regulation of single gene expression but also multi-level regulatory mechanisms underlying the entire sex determination and differentiation process. Consequently, the methylation level differences induced by 5-aza-dC in genes may conceal additional regulatory mechanisms yet to be uncovered. This undoubtedly provides new insights and directions for further exploring the sex regulatory mechanisms in medaka fish.

## 4. Materials and Methods

### 4.1. Experimental Materials and Ethics Statement

The medaka used in this experiment were of the Hd-rR strain (wild-type). The parental medaka was reared in a glass tank with a circulating water system, with water temperature maintained at 26–28 °C, photoperiods of 14L-10D, and fed daily, morning and evening with YYT. All procedures were approved by the Animal Care and Use Committee of Hunan University of Science and Technology, Xiangtan, China. (2023041007) Fish were maintained and conducted in compliance with the ARRIVE guidelines.

### 4.2. Embryos Were Collected and Treated with a 5-aza-dC Gradient Concentration

We carefully selected healthy and vigorous medaka fries with the same genetic background for the experiment. All fish were allowed to spontaneously mate in a spawning pond, after which the fertilized eggs were collected and carefully transferred to Petri dishes containing holf medium, which was replaced daily to ensure optimal conditions. The eggs were incubated at a temperature range of 26–28 °C. At 25 days post-fertilization (dpf), approximately 1200 fries were evenly divided into three groups and treated with 5-aza-dC at concentrations of 0 μg/L (control), 50 μg/L, and 100 μg/L. Each treatment group was replicated three times to ensure the reliability of the results. At 35 dpf, the fries were transferred to glass tanks filled with clear water to minimize the potential effects of high stocking density on gonadal differentiation. Each tank, with a capacity of 5 L, housed 50–55 individuals. The fish were then raised under these conditions until they reached sexual maturity at 120 dpf. At this stage, only healthy and active samples from each of the three treatment groups were selected for further detailed analyses.

### 4.3. DNA Extraction and PCR Amplification

Genomic DNA was extracted from medaka using the Biospin tissue genomic DNA extraction kit. The genomic DNA of medaka was used from PG19/20 as a template and two pairs of PCR, in which PG19/20 was used for the amplification of male-specific fragments, and obB-actin4F/4R was used to test the effect of genomic DNA extraction. The PCR reaction system comprised the following ingredients: total of 20 μL: 2 µL of DNA template, 2 × Taq Mix at 10 µL, 0.5 µL of forward and reverse primers, respectively, ddH_2_O at 7 µL. The PCR reaction procedure was as follows: 94 °C pre-denaturation for 5 min; 94 °C denaturation for 30 s, 57 °C annealing for 30 s, 72 °C extension for 140 s, and a total of 35 cycles; 72 °C extension for 10 min. PCR amplification products were separated by 1% agarose gel electrophoresis at 100 V for 30 min and imaged in a gel imaging system. The sequences of the primers are listed in Table 1.

### 4.4. Paraffin Sections of Gonadal Tissue

Female and male adult medaka treated with 0 μg/L, 50 μg/L, and 100 μg/L of 5-aza-dC were taken, respectively. After collecting the data of the growth and reproduction, the gonadal tissue was fixed in 4% paraformaldehyde at 4 °C for 24 h before dehydration from low to high gradient concentration, and after xylene transparency and paraffin embedding and trimming, routine paraffin serial sections were performed. The sections were then dewaxed for xylene, and the ethanol was water infiltrated by high to low gradient concentration, stained with hematoxylin–eosin, dehydrated again with ethanol at a concentration gradient, and after xylene became transparent, sealed the sheet with neutral gum, and finally observed under a light microscope (Leica Microsystems, Model DMi8 manual, Wetzlar, Germany).

### 4.5. RNA Isolation, cDNA Synthesis, Primer Strategy, and RT-qPCR

35 dpf medaka fries were taken from different treatment concentrations, and total whole fish RNA was extracted by chloroform method, and then genomic cDNA from RNA was removed by treatment with DNase I, and finally RNA was reverse transcribed into cDNA and used as a template for the qPCR reaction. For reverse transcription, RNA was used as a template, and the first strand of cDNA was synthesized by a reverse transcription enzyme. The system was as follows: RNA template of 8 μL, 5 Buffer of 4 μL, dNTP of 4 μL, Oligo dT of 1 μL, 0.5 μL of reverse transcriptase, RNase 0.5 μL of the inhibitor, and 2 μL of the DEPC water. The reverse transcription program was as follows: 42 °C for 30 min; 95 °C for 2 min. In this experiment, *β-actin* was used as the reference gene to detect three DNMTs-related genes (*dnmt1*, *dnmt3bb.1*, and *dnmt3bb.2*) and seven sex-related genes (*cyp19a1*, *foxl2*, *dmrt1*, *sox5*, *amh*, *sox9a*, and *wnt4*) (see Table 2 for primer sequence information). The system used for qPCR was 20 μL (1 µL of cDNA template, 10 µL of SYBR Green Supermix, 1 µL of the forward primer, 1 µL of the reverse primer, and 7 µL of ddH_2_O). The qPCR reaction program was as follows: 95 °C pre-denaturation for 3 min; 95 °C denaturation for 10 s, and 60 °C annealing for 30 s (50 cycles). Melting curve: 65 °C for 5 s and increased by 0.5 °C after each cycle. The qPCR data were collected using Bio-Rad CFX Manager 3.0 software. The relative expression levels of the target genes were calculated using the 2^−ΔΔCT^ method, followed by *t*-test analysis (*p* < 0.05 was considered statistically significant). Finally, the data were visualized using GraphPad Prism 10.1 software.

The transcription ID of the methyltransferase genes (*dnmt1*, *dnmt3bb.1*, and *dnmt3bb.2*) can be searched on https://www.ensembl.org/index.html?redirect=no (accessed on 16 January 2025). Obtain the transcript IDs of genes related to sex differentiation from the NCBI database. The transcription IDs and primer sequences are presented in Table 2.

### 4.6. Bisulfite Sequencing PCR (BSP)

The BSP primers were designed using methyl primer expression v1.0. PCR amplification was performed using the TaKaRa EpiTaq HS kit (Sangong Bioengineering Co., Ltd., Shanghai, China), in a reaction mixture consisting of 1–2 μL of bisulfite-modified DNA, 1 μL of dNTP mixture, 5 μL of 10×EpiTaq PCR buffer, 0.4 μL of EpiTaq (5 U/μL), 1 μL of forward primer, and 1 μL of reverse primer (10 mmol/L). The final volume was adjusted to 50 μL with ddH_2_O. The purified PCR products were cloned into the pMD 18-T vector (Sangong Bioengineering Co., Ltd., Shanghai, China) and transformed into Escherichia coli cells. Ten individual clones were then sequenced for each sample, and the methylation levels of the CpG dinucleotides in the promoter region were assessed. The sequence information of the BSP primers are listed in Table 3.

### 4.7. Statistical Analysis

Graphpad Prism v10.1 was used to analyze all statistical data and make the graphs. Data are presented as the mean ± SEM. Differences between experimental groups were made by an independent sample T-test. The statistical significance was declared at *p* < 0.05, *p* < 0.01, and *p* < 0.001.

## 5. Conclusions

In conclusion, this study elucidated the impact of 5-aza-dC, a DNA methyltransferase inhibitor, on sex differentiation in medaka. Our experimental findings revealed that 5-aza-dC modulates the expression of genes associated with sex differentiation by downregulating the DNA methyltransferase gene, thereby influencing medaka sex differentiation and inducing the transformation of genetic males into physiological females. Through fluorescence quantitative PCR and bisulfite sequencing PCR (BSP), we demonstrated that the expression of DNA methyltransferase genes, *dnmt1* and *dnmt3bb.1*, was inhibited, leading to alterations in the methylation levels of genes related to sex differentiation. Consequently, the expression of male-specific genes, *dmrt1* and *amh*, were suppressed, while the expression of female-specific genes, *foxl2*, *wnt4*, and *cyp19a1*, were upregulated, ultimately resulting in the feminization of medaka. This research provides novel insights into the role of DNA methylation in fish sex differentiation and offers a promising avenue for aquatic applications. These findings hold significant implications for enhancing the efficiency and economic benefits of high-density aquaculture practices.

## Figures and Tables

**Figure 1 ijms-26-03280-f001:**
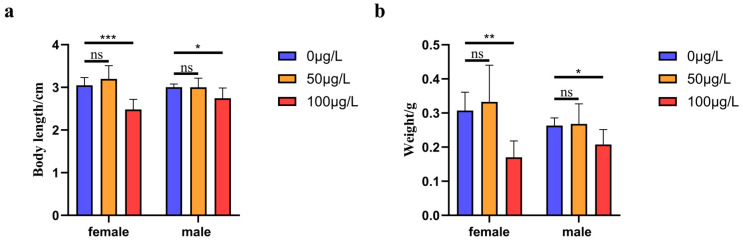
(**a**): Effects of different concentrations of 5-aza-dC treatment on body length of medaka. ns: no significant difference, *: *p* < 0.05, and ***: *p* < 0.001. (**b**): Effects of different concentrations of 5-aza-dC treatment on body weight of medaka. ns: no significant difference, *: *p* < 0.05, **: *p* < 0.01.

**Figure 2 ijms-26-03280-f002:**
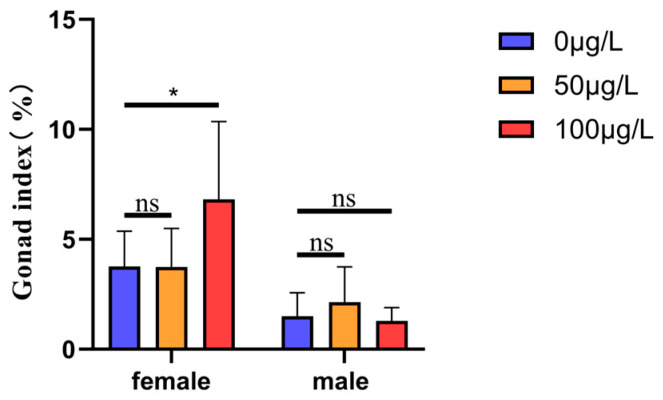
Effects of 5-aza-dC treatment at different concentrations on the gonad index of medaka. ns: no significant differenpce; *: *p* < 0.05.

**Figure 3 ijms-26-03280-f003:**
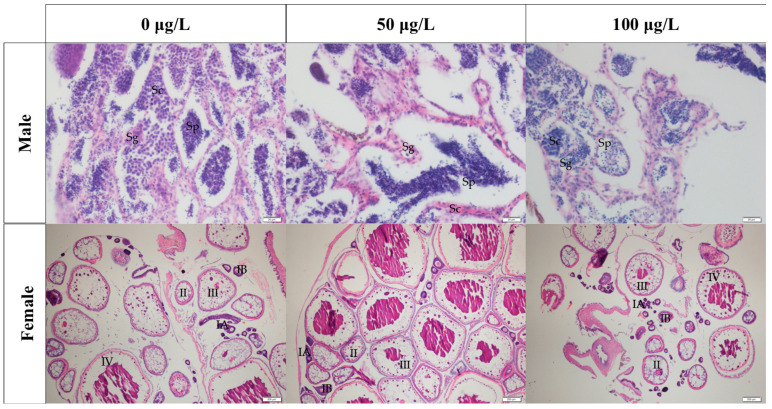
Histological results of gonadal tissue. IA, IB: primary growth stage; Ⅱ: cortical alveolar stage; Ⅲ: vitellogenic stage; IV: oocyte maturation Stage; Sg: spermatogonium; Sc: spermatocyte; Sp: sperm. Microscope’s magnification of male gonadal tissue paraffin section: 4×. Microscope’s magnification of female gonadal tissue paraffin section: 40×.

**Figure 4 ijms-26-03280-f004:**
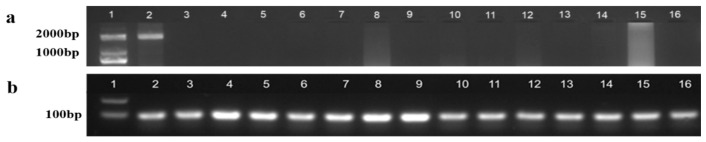
(**a**): Results of PCR amplification of *dmrt1Y.* (**b**): Results of PCR amplification of *β-actin*. 1: DNA marker; 2–16: PCR-amplified product of the sample to be detected.

**Figure 5 ijms-26-03280-f005:**
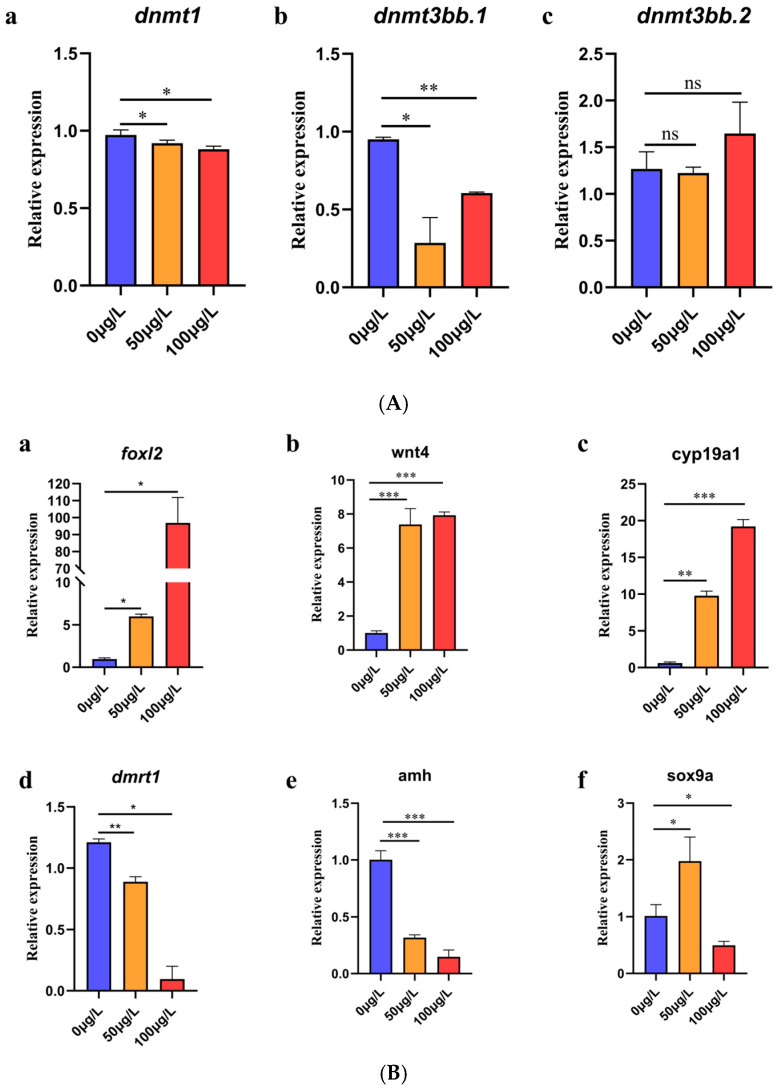
(**A**) a–c: Expression of DNA methyltransferase gene under different concentrations of 5-aza-dC treatment. ns: no significant difference, *: *p* < 0.05, and **: *p* < 0.01. (**B**) a–c: Expression of genes related to female sex differentiation under different concentrations of 5-aza-dC treatment. *: *p* < 0.05, **: *p* < 0.01, and ***: *p* < 0.001. d–f: Expression of genes related to male sex differentiation under different concentrations of 5-aza-dC treatment. *: *p* < 0.05, **: *p* < 0.01, and ***: *p* < 0.001.

**Figure 6 ijms-26-03280-f006:**
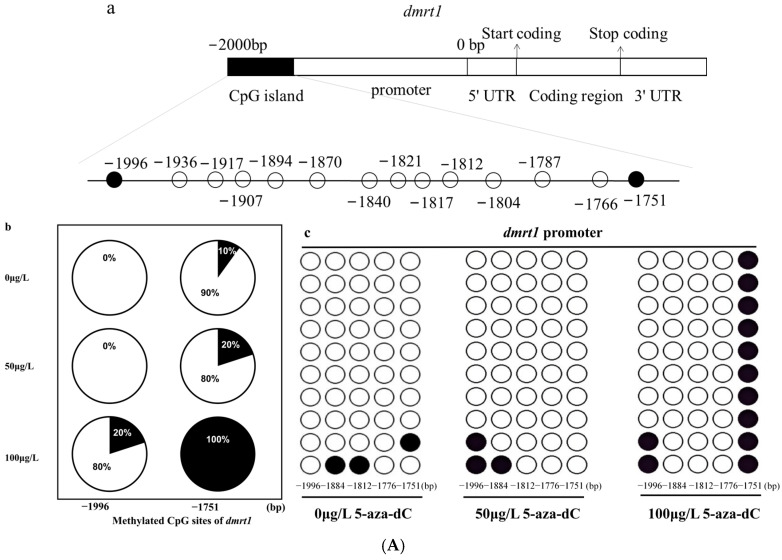
(**A**) 5-aza-dC inhibitors significantly altered DNA methylation levels in *dmrt1* promoters. (**a**): Prediction of CpG island in the *dmrt1* gene promoter. The black areas represent the CpG islands; (**b**): Site-specific DNA methylation of individual CpG in the *dmrt1* promoter. (**b**): The black portion represents the percentage of sequences with cytosine methylation at the anchor site, while the white portion represents the percentage of sequences with unmethylation at the anchor site. (**c**): The methylation patterns of the *dmrt1* promoter. Open and filled circles denote unmethylated and methylated positions, respectively. (**c**): The methylation patterns of the *dmrt1* promoter. Open and filled circles denote unmethylated and methylated positions, respectively. (**B**) 5-aza-dC inhibitors significantly altered the DNA methylation levels of the *foxl2* promoters. (**a**): Prediction of CpG island in the *foxl2* gene promoter. The black areas represent the CpG islands; (**b**): Site-specific DNA methylation of individual CpG in the *foxl2* promoter. (**b**): The black portion represents the percentage of sequences with cytosine methylation at the anchor site, and the white portion represents the percentage of sequences with cytosine unmethylation at the anchor site. (**c**): The methylation patterns of the *foxl2* promoter. Open and filled circles denote unmethylated and methylated positions, respectively.

**Table 1 ijms-26-03280-t001:** The sequence information of the common PCR primers.

Primers	Primer Sequence (5′-3′)	Base Number (bp)	TM (°C)
PG19	GAACCACAGCTTGAAGACCCCGCTGA	26	64.3
PG20	GCATCTGCTGGTACTGCTGGTAGTTG	26	62.8
obB-actin4F	CTCTGGTCGTACCACTGGTATCG	23	61.3
obB-actin4R	GCAGAGCGTAGCCTTCATAGATG	23	59.6

**Table 2 ijms-26-03280-t002:** Sequence information of the qRT-PCR primers.

Gene Name	Transcript ID	Primer Sequence (5′-3′)	Fragment Size (bp)	TM (°C)
*dnmt1*	00000040050	F: AAGAGAAGAAGCGCCTCAAAGT R: TGAAACTCCGCGAAGAAGAGAA	255 bp	61.01/61.01
*dnmt3bb.1*	00000036223	F: ATGACAACAAAGGCTTCTGCAT R: TACGTCTTCGGAAACAACAGTA	261 bp	61.04/60.88
*dnmt3bb.2*	00000036293	F: GGACGAGTACACAGACCACTCC R: CCTCACCAGACACATGAGCAGG	149 bp	61.00/61.02
*foxl2*	001104888.1	F: CCTCGTCCTACAACCCCTACTC R: CTCATCCCCAACATCCTGCTCC	242 bp	61.25/60.93
*wnt4*	001160439.1	F: ACCGCCGATGGAACTGCTCT R: CAGGCCCTTGTGACCGCAAA	216 bp	62.20/61.70
*cyp19a1*	001278879.1	F: CCCTCATCCTGCTCGTCTGT R: AGGACATAAGCGGGCCCAAA	178 bp	59.70/59.70
*dmrt1*	001104680.2	F: AGGAGGAGCTTGGGATTTGTAG R: GATGTTTAGGGTTCGAGGAGGA	265 bp	60.95/60.99
*amh*	001360941.1	F: CTGGCAGAGCAGGAAACGGT R: CACCGTCTTCAGCGCCTTCA	209 bp	60.90/61.00
*sox9a*	001105086.1	F: CGCACGATCCTCAGCAGTCA R: AGGGCGCACAGTCTGATTGA	180 bp	60.50/59.50
*β-actin*	001104808.1	F: AAAAGGGGCTCATTCTCAACTC R: CTCAACTCTTACTCGGGGAAAA	203 bp	60.95/60.99

**Table 3 ijms-26-03280-t003:** Sequence information of the BSP primers.

Primers	Primer Sequence (5′-3′)	Fragment Size (bp)
*foxl2*-1-F	AAAAGTTTATTTAGATGAATATTTTTGA	172 bp
*foxl2*-1-R	AACATATTTATTTTCTACAACCCTACAA	
*foxl2*-2-F	GTATATTGAAGGATGTTGTGTTTTATAT	378 bp
*foxl2*-2-R	CCACCCATATAAATCCCCCT	
*Dmrt1*-F	GTTTGATTTGTATGTATTTGTGTTTAGA	323 bp
*Dmrt1*-R	CTTTCCRATCAAAACRAATACCTT	

## Data Availability

All important data is included in the manusctipt.

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
