# Peer review of "Effects of Treatment with a DNA Methyltransferase Inhibitor 5-aza-dC on Sex Differentiation in Medaka (Oryzias latipes)"

_ijms, 2025, doi:10.3390/ijms26073280_

Round 1
Reviewer 1 Report
Comments and Suggestions for Authors
Title of the Paper: Effects of Treatment with a DNA Methyltransferase Inhibitor 5-aza-dC on Sex Differentiation in Medaka (Oryzias latipes)
Overall Evaluation:
The study investigates the role of DNA methyltransferase inhibitor 5-aza-dC in the sex differentiation of medaka fish by analyzing gene expression, methylation patterns, and physiological effects. The research is relevant and contributes to understanding epigenetic regulation in fish sex differentiation. Additionally, the findings have potential applications in aquaculture, especially in developing sex-controlled breeding techniques, but certain aspects require further clarification and contextualization:
Minor concerns and suggestions:
- Line 126 and 144: What do you mean by “Musklifish”, isn´t your animal model medaka? Please correct accordingly.
- Line 22-23: "5-aza-dC at 0μg/L , 50μg / L and100μg/L"
- Change to: "5-aza-dC at 0 μg/L, 50 μg/L, and 100 μg/L."
- Line 167-168: "The level of gene expression was calibrated by internal reference.The expression"
- Change to: "The level of gene expression was calibrated by an internal reference. The expression"
- Line 62-63: 5-aza-dC is a cytidine analog that knots exclusively with DNA Combined, with little effect on RNA and protein synthesis, and high demethylation efficiency."
- Suggestion: "5-aza-dC is a cytidine analog that binds specifically to DNA, with minimal impact on RNA and protein synthesis, while demonstrating high demethylation efficiency."
- Line 94-97: "The experimental results showed that high concentration of 5-aza-dC; high concentration of 5-aza-dC had a high effect on female gonad index; high concentration of 5-aza-dC increased gene expression level by inhibiting DNA methylation."
- Suggestion: "The experimental results showed that a high concentration of 5-aza-dC significantly affected the female gonad index and increased gene expression levels by inhibiting DNA methylation."
- Line 79-80: "Medaka (Oryzias latipes) is an important animal model,which is easy to breed, large egg laying and short reproduction cycle…"
- Suggestion: "Medaka (Oryzias latipes) is an important animal model, which is easy to breed, lays a large number of eggs, and has a short reproductive cycle."
- Line 52-53: "DNA methylation patterns in the genome achieve mainly catalyzed by maintaining DNA methylation transferases (DNMTs)"
- Suggestion: "DNA methylation patterns in the genome are primarily catalyzed by DNA methyltransferases (DNMTs)."
- Line 215-218: "The black portion represents the percentage of sequences with cytosine methylation at the anchor site, and the white portion represents the percentage of quantas with cytosine unmethylation at the anchor site."
- Suggestion: "The black portion represents the percentage of sequences with cytosine methylation at the anchor site, while the white portion represents the percentage of sequences with unmethylated cytosine at the anchor site."
- Formatting and Citations:
- Ensure consistency in referencing format (some references lack complete details).
Major Concerns:
- Proofread the whole paper; there were a lot of typographical errors and missing spaces between the sentences.
- The justification for choosing the specific concentrations of 5-aza-dC (50 µg/L and 100 µg/L) should be elaborated upon. How were these concentrations determined?
- The rationale behind the timing of treatment (25-35 dpf) should be supported by references or preliminary studies.
- The authors should provide a more detailed statistical methodology, including how they controlled for multiple comparisons.
- Were the sample sizes per group determined based on power calculations? Clarification on sample selection criteria is needed.
- The discussion section should integrate findings with more recent literature on epigenetics in fish sex differentiation.
- The potential off-target effects of 5-aza-dC should be addressed, as global demethylation could have broader implications beyond sex-related genes.
- The biological implications of reduced growth in treated medaka should be discussed further. Is the feminization effect linked to reduced growth, or is it an independent effect?
- The role of DNA methylation in regulating specific sex-determining genes (foxl2, cyp19a1, dmrt1) should be explored in greater detail.
- The impact of 5-aza-dC on other epigenetic mechanisms, such as histone modifications, should be briefly mentioned as a limitation.
- Some figures (e.g., Figure 5) contain a large amount of information but lack clear labeling. It would be helpful to provide more descriptive legends.
- The results of PCR and BSP assays should be quantified and statistically compared.
-
The manuscript contains grammatical errors and awkward phrasing. A professional English language review would improve clarity.
-
Some sections are redundant and could be more concise.
Author Response
Reviewer 1:
Minor concerns and suggestions:
Line 126 and 144: What do you mean by “Musklifish”, isn´t your animal model medaka? Please correct accordingly.
Response: Thank you for your kind advice. We have changed “Musklish” into medaka.(line 101-119)
Line 22-23: "5-aza-dC at 0μg/L , 50μg / L and100μg/L"
Change to: "5-aza-dC at 0 μg/L, 50 μg/L, and 100 μg/L."
Response: Thank you for your advice. As the reviewer has suggested, we have changed it.(line 20-21)
Line 167-168: "The level of gene expression was calibrated by internal reference.The expression"
Change to: "The level of gene expression was calibrated by an internal reference. The expression"
Response: Thank you for your kind advice. As the reviewer has suggested, the sentence has been rewritten.(line 142-143)
Line 62-63: 5-aza-dC is a cytidine analog that knots exclusively with DNA Combined, with little effect on RNA and protein synthesis, and high demethylation efficiency."
Suggestion: "5-aza-dC is a cytidine analog that binds specifically to DNA, with minimal impact on RNA and protein synthesis, while demonstrating high demethylation efficiency."
Response: Thank you for your kind advice. As the reviewer has suggested, the sentence has been rewritten. (line 48-49)
Line 94-97: "The experimental results showed that high concentration of 5-aza-dC; high concentration of 5-aza-dC had a high effect on female gonad index; high concentration of 5-aza-dC increased gene expression level by inhibiting DNA methylation."
Suggestion: "The experimental results showed that a high concentration of 5-aza-dC significantly affected the female gonad index and increased gene expression levels by inhibiting DNA methylation."
Response: Thank you for your kind advice. As the reviewer has suggested, the sentence has been rewritten. (line 71-73)
Line 79-80: "Medaka (Oryzias latipes) is an important animal model,which is easy to breed, large egg laying and short reproduction cycle…"
Suggestion: "Medaka (Oryzias latipes) is an important animal model, which is easy to breed, lays a large number of eggs, and has a short reproductive cycle."
Response: Thank you for your kind advice. As the reviewer has suggested, the sentence has been rewritten. (line 53-54)
Line 52-53: "DNA methylation patterns in the genome achieve mainly catalyzed by maintaining DNA methylation transferases (DNMTs)"
Suggestion: "DNA methylation patterns in the genome are primarily catalyzed by DNA methyltransferases (DNMTs).
Response: Thank you for your kind advice. As the reviewer has suggested, the sentence has been rewritten. (line 43)
Line 215-218: "The black portion represents the percentage of sequences with cytosine methylation at the anchor site, and the white portion represents the percentage of quantas with cytosine unmethylation at the anchor site."
Suggestion: "The black portion represents the percentage of sequences with cytosine methylation at the anchor site, while the white portion represents the percentage of sequences with unmethylated cytosine at the anchor site."
Response: Thank you for your kind advice. As the reviewer has suggested, the sentence has been rewritten. (line 189-194)
Formatting and Citations:
Ensure consistency in referencing format (some references lack complete details).
Response: Thank you for your considerate suggestion. The details of references have been added.
Major Concerns:
Proofread the whole paper; there were a lot of typographical errors and missing spaces between the sentences.
Response: Thank you for your considerate suggestion. We have revised all of these errors.
The justification for choosing the specific concentrations of 5-aza-dC (50 µg/L and 100 µg/L) should be elaborated upon. How were these concentrations determined?
Response: Thank you for your considerate suggestion. We determined this according to the relevant literature, which we have added in the article. (line 70-72 and line 462-464)
The rationale behind the timing of treatment (25-35 dpf) should be supported by references or preliminary studies.
Response: Thank you for your kind advice. We determined this according to the relevant literature, which we have added in the article. (line 70-72 and line 460-461)
The authors should provide a more detailed statistical methodology, including how they controlled for multiple comparisons.
Response: Thank you for your considerate suggestion. We have added the part of statistical analysis. (line 395-399)
Were the sample sizes per group determined based on power calculations? Clarification on sample selection criteria is needed.
Response: Thank you for your king advice. The sample selection criteria has been added. (line 331-333)
The discussion section should integrate findings with more recent literature on epigenetics in fish sex differentiation.
Response: Thank you for your considerate suggestion. The recent literature on epigenetics in fish sex differentiation has been added in the discussion. (line 284-294)
The potential off-target effects of 5-aza-dC should be addressed, as global demethylation could have broader implications beyond sex-related genes.
Response: Thank you for your kind advice. We have added the effect of 5-aza-dc on genes other than sex-related genes. (line 224-227)
The biological implications of reduced growth in treated medaka should be discussed further. Is the feminization effect linked to reduced growth, or is it an independent effect?
Response: Thank you for your considerate suggestion. Our article mainly discusses the issues related to sex differentiation, so the reduced growth didn’t be explained in detail. The feminization effect is an independent effect.
The role of DNA methylation in regulating specific sex-determining genes (foxl2, cyp19a1, dmrt1) should be explored in greater detail.
Response: Thank you for your kind advice. Further research on the role of DNA methylation in regulating sex-determining genes such as foxl2 is ongoing and will be presented in a future article.
The impact of 5-aza-dC on other epigenetic mechanisms, such as histone modifications, should be briefly mentioned as a limitation.
Response: Thank you for your kind advice. The impact of 5-aza-dC on histone modification has been added. (line295-310)
Some figures (e.g., Figure 5) contain a large amount of information but lack clear labeling. It would be helpful to provide more descriptive legends.
Response: Thank you for your kind advice. We have revised the figures, and illustrated this in more detail below Figure 5.
The results of PCR and BSP assays should be quantified and statistically compared.
Response: Thank you for your kind advice. We have revised the results of PCR and BSP assays.
The manuscript contains grammatical errors and awkward phrasing. A professional English language review would improve clarity.
Response: Thank you for your kind advice. We have corrected the grammatical errors throughout the article, and improved the expression on English.
Some sections are redundant and could be more concise.
Response: Thank you for your considerate suggestion. We have checked the whole article and make it more concise.

Reviewer 2 Report
Comments and Suggestions for Authors
follow the attachment

follow the attachment
Author Response
Reviewer 2:
- Very long sentence Line 22-32 and Line : 66-75, affect to understand, make
simple and concise
Response: Thank you for your kind advice. These sentences have become simple and concise. (line 20-26 and line 63-68)
- All gene name should be Italic. Please ensure all gene names are italicized
throughout the manuscript.
Response: Thank you for your advice. All gene name have been Italic.
- Follow uniform reference style. For example in Line 66, Bachtrog et al., 2014. It
you site in number so change it to number
Response: Thank you for your kind advice. The whole article has followed a uniform reference style.
- Line: 79: Oryzias latipes, make it italic
Response: Thank you for your advice. It has been italic.(line 53)
- Line 90-97. I don’t understand, is it introduction or result?
Response: Thank you for your kind advice. It is a summary of result. (line 69-75)
- In some lines, there is no space after a word. It could be an issue with the text
version or the PC. However, you can check this.
Response: Thank you for your kind advice. We have checked the whole article and made it correct.
- The introduction lacks coherence. Refocus it to align with your objective and
clearly address your research statement.
Response: Thank you for your considerate suggestion. We have revised the introduction.
- Is Figure 1a and 1b for Musklifish or Medaka? Be careful if you copied the title
from another manuscript.
Response: Thank you for your kind advice. Figure 1a and 1b are for Medaka. We have changed Musklifish into Medaka.
- Line 139: oryzae, make it italic
Response: Thank you for your advice. It is changed into medaka. (line 114)
- Line 344: light microscope. (Specification required)
Response: Thank you for your kind advice. The specification of light microscope has been added. (line 355-356)
- Line 366: In the qPCR reaction conditions, you consistently used 60°C for all the
selected genes. Were the annealing temperatures the same for all genes? Why
annealing temperature (60°C) and melting (65°C) temp is different? Add the
annealing temperature in table 2
Response: Thank you for your kind advice. We used 60°C for all the selected genes. The difference between annealing temperature and dissolution temperature is due to experimental requirements. The annealing temperature has been added in table 2.
- Can you check the transcription ID in table 2. Is it the NCBI access number of the
selected gene? I didn´t find the assign ID to responsible gene.
Response: Thank you for your kind advice. We found the transcription ID in two websites, and we have added the websites in the end of 4.5 RNA isolation, cDNA synthesis, primer strategy, and RT-qPCR. (line 377-380)
- Methodology is not well describe. Specify how many days the fish were reared
and how often length and weight were measured. How to you calculated gonad
index?
Response: Thank you for your kind advice. We have added all of these in Part 4. (line314-316 and line 320-324 and line 331-333)
- Line 358-364: Revised the sentence, This line is not understandable
Response: Thank you for your advice. The sentence has been revised. (line 368-376)
- Did you follow any animal ethics guidelines for using the fish? Ethical approval
is required.
Response: Thank you for your considerate suggestion. We followed animal ethics guidelines for using the fish. Ethical approval has been added. (line 316-318)
- Ensure consistency in indicating significance levels—use either P or p uniformly
throughout the text.
Response: Thank you for your kind advice. We have changed p into P.
- Overall, the manuscript requires substantial improvement in writing. Seek
support from an English editor if necessary.
Response: Thank you for your advice. We have improved our writing.
- There is no paragraph for conclusions
Response: Thank you for your kind advice. The paragraph for conclusion has been added. (line 400-414)

Round 2
Reviewer 2 Report
Comments and Suggestions for Authors
I have checked all of my comments has been addressed..